# Rapid Mobilization of an Evidence-Based Psychological Intervention for Pediatric Pain during COVID-19: The Development and Deployment of the Comfort Ability^®^ Program Virtual Intervention (CAP-V)

**DOI:** 10.3390/children10091523

**Published:** 2023-09-08

**Authors:** Amy E. Hale, Simona Bujoreanu, Timothy W. LaVigne, Rachael Coakley

**Affiliations:** 1Pain Medicine, Department of Anesthesia, Critical Care and Pain Medicine, Boston Children’s Hospital, Boston, MA 02115, USA; amy.hale@childrens.harvard.edu (A.E.H.); simona.bujoreanu@childrens.harvard.edu (S.B.); timothy.lavigne@childrens.harvard.edu (T.W.L.); 2Division of Gastroenterology, Hepatology & Nutrition, Boston Children’s Hospital, Boston, MA 02115, USA

**Keywords:** adolescents, chronic pain, knowledge mobilization, virtual intervention, cognitive behavioral therapy, telehealth, pediatric

## Abstract

Background: The gold standard of treatment for chronic pain is a multidisciplinary approach in which psychology plays a leading role, but many children and caregivers do not gain access to this treatment. The Comfort Ability^®^ Program (CAP) developed a CBT-oriented group intervention for adolescents and caregivers designed expressly to address access to evidence-based psychological care for pediatric chronic pain. Before the COVID-19 disruption of in-person services, the CAP workshop had been disseminated to a network of 21 children’s hospitals across three countries. In March 2020, a virtual (telehealth) format was needed to ensure that children with chronic pain could continue to access this clinical service throughout the CAP Network. Methods: A model of knowledge mobilization was used to adapt the CAP workshop to a virtual format (CAP-V) and disseminate it to network sites. A pilot study assessing participant and clinician perceptions of acceptability, feasibility, and treatment satisfaction included baseline, post-sessions, and post-program questionnaires. Results: A knowledge mobilization framework informed the rapid development, refinement, and mobilization of CAP-V. Data from a pilot study demonstrated feasibility and high acceptability across participants and clinicians. Conclusions: A knowledge mobilizationframework provided a roadmap to successfully develop and deploy a virtual behavioral health intervention for adolescents with chronic pain and their caregivers during a worldwide pandemic. While CAP-V has demonstrated preliminary clinical feasibility and acceptability at the CAP hub, ongoing research is needed.

## 1. Introduction 

Chronic pediatric pain is both common and expensive, placing a significant burden on healthcare systems. With a prevalence rate of 25% to 46% [1,2], it ranks among the most expensive pediatric health problems in the United States, costing an estimated $19.5 billion dollars annually [3]. Recent analyses of Canada’s healthcare systems revealed that the cost of chronic pain in 2019 was found to be $40.3 billion, with an estimated additional cost increase of 36.2% by 2030 [4]. Pediatric pain is complex both in presentation and impact: of children who experience chronic pain, 29% have more than one kind of pain, 80% are at risk for pain into adulthood, 95% report problems with school, and 70% have problems with sleep [5,6]. Moreover, mental health disorders such as anxiety and depression are up to three times more prevalent in children and adolescents with chronic pain [7,8]. Within the United States, there is also widespread recognition that chronic pain has contributed to a national epidemic of opioid abuse and dependence [9,10], with an especially worrisome trajectory for adolescents [11,12]. Timely access to treatment is essential since untreated chronic pain may persist or worsen over time and impact children’s physical, emotional, and social development [13]. Given the current scope of pediatric pain, there is presently a global effort to mobilize targeted clinical services.

According to 2020 guidelines by the World Health Organization, a multidisciplinary approach—inclusive of behavioral health intervention—is best practice for the treatment of pediatric pain [14]. These guidelines reflect decades of research demonstrating that cognitive behavioral therapy (CBT), the most-researched behavioral health treatment modality, reduces the severity and prevalence of chronic pain, pain-related disability, and pain-related comorbidities such as depression and anxiety [15,16]. The involvement of caregivers in pain-focused behavioral health intervention is also crucial since caregivers often report feeling unequipped to support their child’s recovery [17]. Importantly, research shows that when caregivers are provided with evidence-based CBT education and skills training, there is both a reduction in caregiver stress and improved child outcomes [18,19]. 

Despite clarity regarding best practices for pediatric pain treatment, there remain substantial gaps in clinical care when it comes to behavioral health access for pediatric patients with chronic pain. Part of the problem stems from the lack of connectivity between evidence producers (researchers) and evidence consumers (institutions, clinicians, and patients) [20]. Most clinical interventions are researched and developed with high internal reliability and rigid implementation structures to minimize confounding results and determine efficacy. While this is an imperative first step, a systematic focus on external validity and implementation science is needed to broadly shift clinical care. Without attention to the various real-world needs of complex and overburdened healthcare systems, it is difficult to embed and sustain new interventions or clinical care pathways. As a result, within pediatric chronic pain, many families do not reliably gain access to evidence-based behavioral healthcare [21,22,23,24,25]. Within the field of knowledge mobilization, the theoretical Knowledge-to-Action (KTA) model developed by Graham et al. [26] illustrates how it is possible to generate and disseminate interventions with high external validity, meaningfully helping to close the knowledge-to-practice gap. For a thorough review of this KTA framework within pediatric pain treatment, please refer to Coakley and Bujoreanu 2020 [27].

The Comfort Ability^®^ Program (CAP) has developed a collection of interventions and assets focused on reducing the multiplicity of producer-, provider-, institutional- and patient-level barriers to accessing behavioral health interventions for pediatric chronic pain. The first CAP intervention designed was a 1-day, in-person workshop for children and adolescents with chronic pain (aged 10–17 years old) and their caregivers. This in-person workshop (hereafter referred to as CAP-IP) includes a total of twelve hours of intervention delivered in parallel (six hours of intervention for youth with chronic pain and six for their caregivers), in an intensive workshop format. In keeping with the psychology evidence base in pediatric pain, the core components of CAP-IP include psychoeducation, pain neuroscience education, hands-on skills training, and social support. The intervention’s condensed 1-day workshop format affords families the opportunity to receive a full “dose” of treatment in a single day as opposed to making multiple trips to the clinic for treatment.

In March 2020, CAP-IP had been running continuously at the CAP hub children’s hospital for 9 years andhad been modified to create a new intervention specifically for patients with sickle cell disease-related pain (CAP-SCP [28]).These two programs were disseminated to an additional 20 institutions across the United States, Canada, and Australia. The KTA framework [26] was used to guide CAP’scross-institutional spread, with core priorities of maintaining intervention fidelity (ensuring consistent outcomes), flexibility (facilitating adoption across diverse institutions and clinicians), and sustainability [27]. Working within the KTA dissemination framework, CAP formed a collaborative network of sites (hereafter the CAP Network) that subsequently helped to identify additional CAP-related knowledge-to-practice gaps and to co-create additional patient- and provider-facing resources.

It was in this context that the COVID-19 pandemic interrupted clinical operations across the world. When CAP-IP workshops were halted throughout the CAP Network for the foreseeable future, there were significant concerns about access to psychological care for pediatric patients with chronic pain. The already-long waitlists for behavioral health services were anticipated to grow with the disruption to face-to-face clinical care. Compounding this, there was an anticipated increase in pediatric chronic pain rates overall (both new diagnoses and relapses) as a result of the pandemic. As institutions and clinicians scrambled to shift clinical operations, it became clear that the virtual/telehealth landscape did not offer any parallel interventions that would meet the clinical needs of both patients and their caregivers in the way that CAP-IP had done. Given the extraordinary context and concerns of the COVID-19 pandemic, a decision was made to adapt the original CAP-IP workshop to an entirely virtual telehealth format (hereafter referred to as CAP-V) using Graham et al.’s KTA framework [26].

## 2. Development of CAP-V

As with the initial CAP-IP development process, CAP-V first emerged in the context of the identification of an access barrier (COVID-19 restrictions on in-person clinical care) and the need for an evidence-based psychological intervention that could be delivered virtually and disseminated through the growing CAP Network. Building on the successful development of CAP-IP and CAP-SCP [27], CAP-V was developed via a similar iterative process inclusive of three overarching stages: (1) generation and refinement of program assets, (2) dissemination, and (3) maintenance and innovation (Figure 1). These phases were iterative such that although asset generation was the starting point, knowledge gained through the dissemination and maintenance phases subsequently informed further asset generation.

### 2.1. Generation and Refinement of Clinical Assets

The first stage of CAP-V development centered on transitioning the in-person workshop into a viable virtual format with a focus on providing the same clinical content, social support, skills practice, and overall implementation structure and procedures. Careful consideration was given to the factors related to using a virtual interface in a way that invites developmentally congruent participant engagement and contributions while recognizing the needs of the participants (children and adolescents struggling with pain conditions that impact stamina, focus, and ability to engage with computer screens). Across both the adolescent and caregiver interventions, it was important to create appealing and interactive materials and include activities that would foster social connection and community through a telehealth interface.

In addition to generating new clinical content, the development of CAP-V also required the generation of clinical and systems-level implementation procedures. The prospect of having both adolescents and caregivers participate in a virtual program from their homes over the course of a full day at the same time presented both technical challenges and a high probability of “screen fatigue”. Therefore, one of the most significant systems-level changes between CAP-IP to CAP-V was the shift from a 1-day workshop to a multi-session intervention. The caregiver intervention was divided into two extended sessions (180 min each) delivered over a weekend, while the adolescent intervention was divided into four sessions (90–120 min each), delivered across two consecutive weeks. In addition to the new administration structure, new implementation procedures were needed to ensure safety in the virtual group setting. CAP-V included standardized procedures to verify adolescents’ locations during each session, confirm contact information for the adolescent and a caregiver, and plan for technical troubleshooting, as needed. Beyond implementation procedures that informed direct clinical care, assets were also developed for clinician support around clinical implementation, telehealth interface use, and technical troubleshooting.

### 2.2. Intervention Refinement

In April 2020, 1 month after the COVID-19 pandemic stopped all outpatient in-person clinical operations at the CAP hub institution, the new CAP-V intervention was clinically administered for the first time. This initiated an iterative 4-month process of asset generation and refinement within the CAP Network. In addition to the CAP hub institution, two CAP Network sites clinically administered pilot versions of CAP-V. Between April and July 2020, CAP-V was implemented 8 times: 4 times at the CAP hub institution and 4 times across two sites within the CAP Network. After each of these CAP-V administrations, refinements were made to the content and structure of the CAP-V clinical and systems-level assets.

### 2.3. Dissemination

In October 2020, after a 2-month period of asset refinement (August–September 2020), the CAP hub began a soft roll out to sites within the CAP Network who wanted to adopt CAP-V. Training was offered via a live virtual group format and focused on clinical content, systems-level implementation, and guidance on troubleshooting technological challenges for virtual groups.

By January 2021, CAP-V had been successfully disseminated, implemented, and monitored for fidelity at seven sites. Additional sites within the CAP Network were seeking training and new institutions were also interested in adopting the intervention. To continue to rapidly train clinicians and transfer the intervention to new sites with fidelity, a more streamlined dissemination process was needed. New training assets were developed and deployed through a professional training website developed for CAP (CAP-Pro) [27]. Materials included downloadable administration manuals, patient recruitment materials, instructional on-demand training videos, research resources (grant writing, quality improvement assessments), and a complete administration of a mock CAP-V intervention for training purposes (both adolescent and caregiver groups). While all sites within the CAP Network also had multiple 1:1 virtual consultations and fidelity monitoring with a CAP hub trainer, the asynchronous learning afforded through the CAP-Pro central training site allowed for more efficient dissemination.

### 2.4. Maintenance and Innovation

Following the initial generation and dissemination of CAP-V, the focus shifted to maintenance and innovation. Through this important stage of monitoring knowledge use, additional clinical and systems-level challenges within the CAP Network were identified. For example, due to the healthcare crisis caused by the pandemic, some CAP Network sites were recruiting clinicians with limited pediatric chronic pain-specific experience to help co-lead CAP-V. To address the variation in clinician training, CAP created a 4-hr continuing-education credit course (Foundations of Pediatric Pain Management) to broadly teach about research-based behavioral health interventions in pediatric chronic pain, giving greater context to the specific goals of the CAP-V intervention. This on-demand course was deployed on CAP-Pro and therefore accessible to all clinical, research, and administrative staff within the CAP Network.

## 3. Methods

### 3.1. Study Design

Following the initial development and refinement of CAP-V, a single-arm study was designed to examine the intervention’s feasibility and acceptability at the CAP hub site. The study was approved by the institutional IRB and recruitment for this study occurred from October 2020 until February 2022. Study inclusion criteria were English-speaking adolescents aged 10–17 years with chronic pain and their caregivers attending CAP-V. Caregivers were introduced to the study at the time of CAP-V registration and those who expressed interest in the study were sent a secure link via REDCap [29,30] to complete informed consent and baseline measures prior to attending the intervention. Participating caregivers from two-caregiver households identified one as a “primary responder” vs. a “secondary responder” to ensure accurate assessment at both individual and family levels across timepoints. The current study represents a part of a larger study with one additional timepoint, and as part of the larger study procedures, participants were given $20 gift cards per family for the completion of all timepoints.

To reduce healthcare disparity and patient barriers in accessing telehealth services, the CAP hub site offered all participants a variety of options including free hot spots, loaner tablets or laptops, and the option of coming to the hospital to complete CAP-V telehealth visits in a private room with computer access.

### 3.2. Assessments and Measures

Quantitative and qualitative data related to program recruitment, participant self-report, and clinician experiences were gathered at baseline, after each individual session, and immediately post-intervention (Table 1). A record of patient-reported technology requests (e.g., need for a hot spot, loaner screen, or on-site access to a computer) was collected at the time of CAP-V registration. Clinicians completed a technology experience survey post-intervention.

Intake and Demographics Survey: Caregivers provided demographic and pain-related information for their child at baseline including the child’s age, presenting pain problem(s), referral source, caregiver education, caregiver marital status, child treatment history (e.g., previous use of medication, participation in physical therapy, cognitive behavioral therapy, other treatment modalities, etc.), and caregiver goals for participation in CAP-V.

Technology Experience Survey-Participant: To track challenges related to the virtual/telehealth format of the intervention, participants rated their technology experience including ease of access to the secure links to the CAP-V session, using the telehealth interface, audio or video disruptions, and issues related to Wi-Fi connectivity (lag, disrupted service, dropped signals). Responses included 0 (not present), 1 (minimal disruption, <5 min delay), 2 (moderate disruption, 6–10 min delay), or 3 (significant disruption, >11 min delay).

Telepresence Survey: This measure was designed for this study to evaluate the impact of technology on the group experiences. Participant responses were assessed on a scale of 1 (strongly disagree) to 5 (strongly agree) and items focused on the experience of the telehealth intervention such as privacy protection, preference for telehealth over in-person care, and a sense of support. This measure also included an open-ended item asking for participant suggestions to improve the experience.

Abbreviated Acceptability Rating Profile (AARP): This measure, adapted with permission from the authors, assesses pain treatment acceptance in pediatric patients with pain [31]. Participants completed this measure after the final CAP-V session. Responses were scored from 1 (strongly disagree) to 5 (strongly agree) with a summed score reflecting the acceptability of CAP-V as treatment.

Post-Session Satisfaction: Adolescents and caregivers completed a brief survey designed for this study after each CAP-V session, with response ratings of 1 (strongly disagree) to 5 (strongly agree) focused on participants’ sense of comfort and clinical support during each session. Sample questions asked whether participants felt they could easily ask questions and discuss topics that were important to them during that session, if the information presented in that session was relevant to them, and their perceived quality of the care within that session. Acceptability of the length and structure of each session was also evaluated.

Post-Treatment Satisfaction Survey: Originally designed for CAP-IP [32] and adapted for CAP-V, this survey assesses whether participants believed the CAP-V treatment objectives were met and their satisfaction with the intervention as a whole. Satisfaction was assessed on a scale of 1 (strongly disagree) to 5 (strongly agree) with participants rating items including the extent to which they believed the virtual platform worked well, whether there was enough time to ask questions, if they intended to make changes regarding pain management after the program, their experience of support from clinical group leaders, and whether they would recommend the program to peers. Participants also reported their preferences regarding the program format and had the option of providing open-ended feedback as well.

### 3.3. Clinician Measures

Technology Experience Survey- Clinician: In addition to information collected from individual participants, clinicians indicated how many total participants in each group were observed to have a technology disruption (e.g., audio or video lag, loss of connectivity) and whether or not the clinician also had this issue in each session.

## 4. Results

### 4.1. Demographics

From the baseline enrollment of adolescents (*n* = 73) and caregivers (*n* = 120), 89% (*n* = 68) of adolescents and 86.7% (*n* = 104) of caregivers completed the post-session and post-treatment measures. Adolescent participants had a mean age of 14.6 years (SD 2.1 years), 83.8% (*n* = 62) were assigned female at birth, 78.4% were Caucasian (*n* = 58), and 9.5% were multiracial. Most adolescents (86.5%, *n* = 64) were referred to CAP-V by a provider at the urban tertiary hospital: 55.4% (*n* = 41) came from the pain clinic, 17.6% (*n* = 13) from gastroenterology, 8.1% (*n* = 6) from neurology, and 4.1% (*n* = 3) from rheumatology. The median reported pain duration was 1–2 years. The primary pain type was abdominal (29.7%, *n* = 22) followed by headache (25.7%, *n* = 19), musculoskeletal (17.6%, *n* = 13), neuropathic (16.2%, *n* = 12), and diffuse/widespread pain (8.1%, *n* = 6). At the time of their participation in CAP, 34% of adolescents (*n* = 45.9) were currently enrolled in some form of CBT (not necessarily pain-focused) and 12.2% (*n* = 9) had participated in the past. Of participating caregivers, 94.5% (*n* = 69) were mothers, 77% (*n* = 57) were from two-caregiver households, and 56.8% (*n* = 42) were attending CAP-V with another caregiver. Of the primary parent responders, 41.9% (*N* = 31) had a 4-year college degree and 39.7% (*n* = 29) had a graduate degree; they reported that additional caregivers were similarly well-educated such that 28.4% (*n* = 21) had a 4-year college degree and 37.9% (*n* = 28) had a graduate degree.

Of note, the percentage of participants (adolescents and caregivers) who completed all measures at all time points varied. Hence, the data reported reflect percentages for each individual construct based on the sample that completed the item or measure. Retention rates were greater than 77% for adolescents and 65% for caregivers across all time points.

### 4.2. CAP-V Feasibility

A key component of feasibility for a virtual intervention was determining whether technology barriers may influence who can participate in the intervention and understanding the unique challenges inherent in technology-dependent care.

Of the approximately 200 families who completed an intake for participation in CAP-V during the time of recruitment, only one family reported a potential technology barrier and opted to come to the hospital to participate in CAP-V telehealth sessions in a private room with computer access. No families requested access to a free Wi-Fi hotspot or loaner screen.

Participant-rated technology experience was overall very positive. Adolescents and caregivers collectively reported on a total of 420 virtual sessions. For most sessions (81%), there were no reported technology difficulties. There were small technology difficulties (resulting in 1–5 min of delay) for video connectivity (loss of video or video lag) in 16% of sessions and/or audio connectivity (loss of audio or audio lag) in 14.4% of sessions. There were a few technical difficulties (lack of audio connectivity, difficulty accessing the secure link for the session) that caused significant delays (>11 min), but these occurred in less than 2% of sessions.

The clinician-rated technology experience, assessed across a total of 126 sessions, was similarly very positive. The most common technical problem reported by clinicians was loss of video/video lag, which occurred in 5.6% of sessions. Clinicians also reported minor problems accessing the secure link/logging into the session and audio connectivity issues (loss of audio or audio lag), both of which only occurred 4% of the time.

### 4.3. CAP-V Acceptability

CAP-V acceptability was viewed through three lenses: (1) understanding how participants felt CAP-V addressed their concerns about chronic pain and was felt to be a relevant and meaningful intervention, (2) how well CAP-V mirrored the core elements of CAP-IP, namely psychoeducation, pain neuroscience education, skills training, and social support, and (3) participants’ overall satisfaction with the CAP-V intervention.

Ratings on the AARP across both groups suggest that adolescents and caregivers felt that CAP-V addressed their concerns about chronic pain and was a relevant intervention. For example, in response to the statement “I would be willing to use this treatment for my pain”, 91.2% (*n* = 52) of adolescents responded with “agree” or “strongly agree”. Similarly, caregivers reported positive perceptions of the treatment with 86.9% (*n* = 79) rating “agree” or “strongly agree” to the statement “This is an acceptable treatment for my child’s pain”, and 96.7% (*n* = 88) indicating agreement with, “I would be willing to use this treatment with my child”. Caregivers were also generally optimistic about the potential effectiveness of the intervention, with 83.1% (*n* = 74) reporting their agreement with “This treatment is a good way to handle my child’s pain” and 88.9% (*n* = 80) agreeing with “Overall, this treatment will help my child”. Adolescents were a little less optimistic, with just 72% (*n* = 42) of adolescents agreeing with the statement, “This treatment is a good way to handle my pain”, and 72.4% (*n* = 43) reporting agreement with, “Overall this treatment will help me”. Of note, where adolescents reported lower agreement, there was mostly ambivalence (endorsement of “unsure”) rather than disagreement.

Adolescent and caregiver reports also indicated that CAP-V mirrored the core elements of CAP-IP. Adolescents reported confidence about the material learned, with 98.3% (*n* = 57) choosing “agree” or “strongly agree” as to whether they learned about the mind-body connection and similarly high responses for pain neuroscience education (e.g., how pain works in the body) and skills acquisition (e.g., how to do diaphragmatic breathing, how active distractions can help reduce focus on pain). Moreover, all adolescent participants (*n* = 59) reported that they learned core CBT concepts (e.g., how thoughts and feelings can make pain better or worse) and 93.1% (*n* = 54) reported success in making a unique Comfort Ability^®^ plan. Caregivers also endorsed learning key content areas (“agree” or “strongly agree”) at 94.5–98.9% of the time on almost all content domains (e.g., “In this program I learned about cognitive behavioral therapy”, “In this program I learned the difference between acute and chronic pain”, “In this program I learned new ideas for how to respond to my child’s pain”, and “In this program I learned about the importance of keeping focus on function to best support my child”). The only exception to caregiver knowledge acquisition was in the role of child and adolescent development in pediatric chronic pain, of which 19.8% (*n* = 18) reported that they were unsure about learning. From a social support perspective, both caregivers and adolescents reported positively, but percentages were overall somewhat lower in this domain than in areas of knowledge. Most adolescents 91.4% (*n* = 53) reported that they found it helpful to talk to peers with pain and 96% (*n* = 57) felt there was enough time to ask questions and talk about things that were important to them. Similarly, 83.1% (*n* = 74) of caregivers felt they had enough time to ask questions and discuss matters that were important to them and 86.8% (*n* = 79) felt the time talking and sharing with other caregivers was productive.

Regarding overall intervention satisfaction, adolescents were generally very satisfied. For example, 91.5% (*n* = 54) of adolescents reported they enjoyed CAP-V activities, 79.3% (*n* = 46) stated that they had more confidence in managing their pain after the intervention and 94.8% (*n* = 55) reported that they planned to make changes to their routines to help them feel better. Most (91.4%, *n* = 53) also reported being glad that they did the CAP-V intervention. Caregivers were also generally pleased, with 91.1% (*n* = 82) of caregivers indicating that CAP-V was successful and 94.5% (*n* = 86) reporting that they planned to make changes in how they handled their child’s pain after completing the intervention. Most caregivers endorsed the current structure of CAP-V (2 sessions, 180 min each), but there was some variability. Over half of the caregivers (62.6%, *n* = 57) thought the number of sessions in the virtual intervention was “just right” and 28.6% (*n* = 26) felt there were not enough sessions. Similarly, 60.7% (*n* = 54) thought the length of each session was “just right” while 15.7% (*n* = 14) thought they were too short and 20.2% (*n* = 18) thought they were “a little long”. Despite the variability, only 39.5% of caregivers (*n* = 34) indicated they would have preferred 3 sessions of 120 min each, and only 17.9% (*n* = 14) indicated they would have preferred 4 sessions of 90 min each. See Table 2 and Table 3 for complete acceptability results.

### 4.4. Telepresence Outcomes

Adolescents were generally positive about their telehealth experience. Most reported being willing to use a virtual approach again (71.9%, *n* = 41) and felt it was easy to participate in the virtual session (87.9%, *n* = 51). However, only 37.3% (*n* = 21) of adolescents reported a preference for virtual sessions (over attending an intervention in person), an opinion that did not change whether adolescents attended CAP-V early or late in the pandemic. Despite preference for an in-person experience, adolescents did report feeling supported in the virtual sessions (91.5%, *n* = 54) and felt they were able to support others (79.7%, *n* = 47). Caregivers were more uniformly positive about the virtual experience. Most (93.3%, *n* = 83) had been involved in a previous telehealth visit prior to the intervention, and 90.1% (*n* = 82) reported willingness to use a virtual approach again. Notably, most caregivers (74.7%, *n* = 68) felt that participating in virtual sessions was better than attending in-person sessions. Caregivers also felt that they both felt supported (93.4%, *n* = 85) and were able to support other participants (80.7%, *n* = 75) during the virtual intervention. Qualitative feedback regarding the telehealth experience from all participants highlighted the importance of breaks and appreciation for the physical resource of a printed workbook as a companion guide throughout the virtual intervention.

## 5. Discussion

Evidence-based psychological treatment for pediatric chronic pain is now considered a standard of care, but there remain significant clinical and systemic gaps that frequently impede access to this care. A brief, intensive, CBT-based workshop for adolescents with chronic pain and their caregivers (CAP-IP) was designed expressly to address these gaps in care and has shown to be beneficial to adolescents living with chronic pain and their caregivers [32]. The COVID-19 pandemic disrupted face-to-face clinical care throughout the world—and through the CAP Network—and exacerbated the already-growing epidemic of pediatric chronic pain and the adolescent mental health crisis. While there existed a handful of well-designed apps and internet-based behavioral health treatments targeting pediatric chronic pain at the time the pandemic began [33,34,35], there were no virtual treatments that were comprehensive enough to replace the clinical behavioral healthcare that was previously provided by CAP-IP.

A modified telehealth intervention for adolescents with chronic pain and their caregivers (CAP-V) was generated, refined, and disseminated using the KTA model, a knowledge mobilization framework that guides responsive and adaptive changes to rapidly bridge the gap between knowledge producers and clinical care [26]. The development of CAP-V included clinical materials, a standardized model for training and transferring knowledge to CAP Network sites, provision of supervision, fidelity monitoring, and consultation on program management.

Sample characteristics including age, presenting pain problem, and pain duration in this single-arm CAP-V feasibility study were consistent with previous CAP-IP studies, suggesting that CAP-V targeted the same clinical population as CAP-IP. Results from the study suggest that the CAP-V intervention is both feasible and acceptable to adolescents and caregivers.

Notably, the pivot from CAP-IP to the virtual/telehealth-based intervention (CAP-V) was anticipated to present challenges to the core clinical CAP elements, including reduced educational engagement (psychoeducation and pain neuroscience education), reduced confidence in skill acquisition, and loss of social connection. Encouragingly, this did not seem to be true for most of the quantitative and qualitative reports. Although this study did not determine statistical congruence between CAP-IP and CAP-V, adolescent and caregiver reports of clinical knowledge and skills training and perceived improvements in the capacity to manage pain were very positive and similar to previous reports of CAP-IP [32,36,37]. One notable finding is that CAP-V caregiver reports were lower for the satisfaction item that rated the connection between adolescent development and chronic pain. This rating was consistent with a previous study of CAP-IP [32], and served to inform a subsequent workshop modification eliminating this section from the intervention.

Although there were slightly lower acceptability ratings for social support (adolescent and caregiver) in CAP-V, these remained positive overall. Given that COVID-19 precluded the opportunity for in-person group care for almost 2 years, CAP-V offered essential social support for struggling patients and families, but understandably, many adolescents indicated a preference for in-person care. In contrast, caregivers were highly satisfied with the virtual format. Reasons for this preference were not assessed directly, but it may have been that caregivers appreciated the convenience of accessing the intervention without travel to the hospital, while adolescents may have had more focus purely on the individual experience of virtual vs. in-person social interactions.

Importantly, technology access and use (access to technology, ease of telehealth interface) did not present a significant barrier to engagement in CAP-V sessions. Although all adolescents and caregivers were offered free technology support (Wi-Fi hotspots, access to loaner screens, option of an on-site computer room), only one family took advantage of this option. This unanticipated finding is likely the result of several factors. First, the sample in this study was highly educated, suggesting a higher SES cohort that may have had greater access to technology. Additionally, the study was conducted at a large urban tertiary hospital at the height of the COVID-19 pandemic. Schools in the area had already disseminated computers and hotspots to almost all families with school-age children. Additionally, almost all families in the study had already completed at least one telehealth visit prior to participating in this intervention. Thus, most families had both access to technology and comfort in using a telehealth interface. It is likely that if this study had been completed prior to the pandemic, access to technology and confidence in using a telehealth interface would have been different.

While most CAP-V sessions were not at all disrupted by technology issues (access to secure links, loss or lag of audio or video), small disruptions were present in approximately 18% of the CAP-V sessions and lengthy disruptions were present in approximately 2% of sessions. As a result of these disruptions, CAP-V clinicians occasionally had to meet with an adolescent or caregiver individually to ensure they did not miss core content due to these glitches. However, clinician report of technology experience suggests that, in general, the technology associated with virtual visits was not believed to impact clinical care. Also of note, there were no adverse events (e.g., safety concerns) reported throughout the entire study. This is in line with the previous findings about CAP-IP [32] but also suggests that the safety check procedure at the beginning of each session (confirming contact information for an adolescent and a caregiver to be used in the event of a disconnection or safety concern) was adequate.

Interestingly, the changes in the CAP-V’s clinical structure brought about several potential benefits. First, the extension of the adolescent program across four sessions vs. CAP-IP’s single-day program created an opportunity for participants to independently practice the skills learned during the session, followed by opportunities for group discussion about individual experiences at the next scheduled session. Additional research will be needed to assess the potential benefit of “homework” and process time on skills acquisition. Secondly, the shift to a virtual format allowed adolescents and caregivers to access this program regardless of where in the state they were located, minimizing attendance difficulties that are related to transportation. Third, delivering CAP virtually to caregivers increased the rates of two caregivers participating in the intervention if it was a two-caregiver household. Further research will be needed to determine if this was an artifact of the pandemic (e.g., caregivers staying home due to COVID-19 were more available for the intervention ) or if it was related to the convenience of telehealth services more broadly. As the critical phase of the pandemic has passed and normative clinical operations have resumed, CAP-IP workshops have been reinstated throughout the CAP Network. However, many CAP Network sites (including the CAP hub) continue to offer CAP-V as an intervention option, improving access to care and affording adolescents and their caregivers some choice in how they would like to obtain psychological intervention for chronic symptoms. Thus, there are continued opportunities to study CAP-V outcomes, evaluating potential benefits and drawbacks.

## 6. Limitations and Future Directions

The results of this pilot study must be interpreted within the context of its significant limitations. First, this was a single-arm, cross-sectional study design and results cannot be generalized. A more comprehensive, longitudinal, multi-site study assessing CAP-V is presently underway. Additionally, this study was conducted at an urban tertiary hospital with a sample of well-educated families. These factors may have biased the telepresence and technology experience questionnaires, artificially reducing barriers to access. Within this pilot study, there was also some variability regarding participant completion of assessments at various time points, thereby limiting the conclusions that can be drawn from the data. Thus, future studies evaluating the impact of CAP-V on its own and compared to CAP-IP are needed.

## 7. Conclusions

The rapid development and deployment of CAP-V during the COVID-19 pandemic, a process that involved the iterative generation and refinement of intervention assets, training materials, and a dissemination plan through the CAP-Network, illustrates how the KTA framework can guide clinical innovation. Based on an existing, well-established intervention (CAP-IP), the KTA cycle provided a roadmap to successfully develop and deploy a behavioral health intervention that could be successfully integrated into the clinical care pathway for adolescents with chronic pain. During the worldwide COVID-19 pandemic, when access to behavioral care was severely limited and chronic pain rates were increasing, CAP-V provided access to an essential component of pediatric chronic pain care to many adolescents and caregivers. CAP-V has demonstrated clinical viability and acceptability at the CAP hub. Additional research is needed to determine feasibility, acceptability, and efficacy broadly in the CAP Network.

## Figures and Tables

**Figure 1 children-10-01523-f001:**
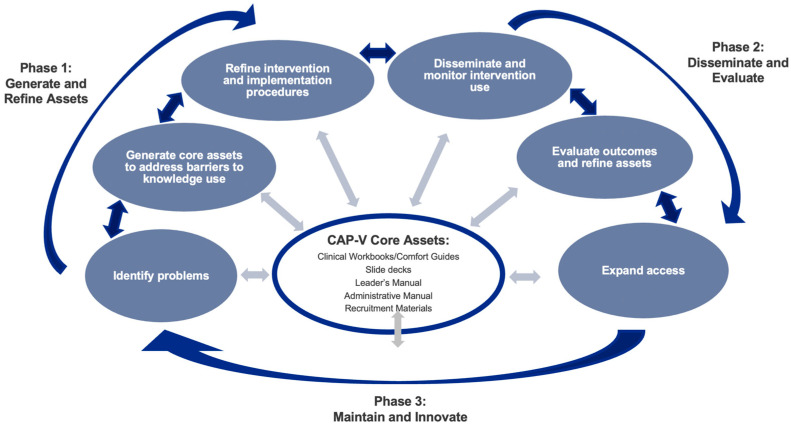
Comfort Ability^®^ Program (CAP-V) development process. Adapted from Graham et al. (2006) [26]; Coakley and Bujoreanu (2020) [27].

**Table 1 children-10-01523-t001:** Measures completed * and study timeline.

Measures	Pre-Treatment	Post-Session	Post-Treatment
Intake and Demographics Survey	C		
Telepresence Survey			A/C
Technology Experience Survey		A/C/L	
Abbreviated Acceptability Rating Profile			A/C
Treatment Satisfaction Surveys		A/C	A/C

* Completion by clinician group leaders (L), caregivers (C), and adolescents (A).

**Table 2 children-10-01523-t002:** Adolescent Post-Treatment Satisfaction Survey.

	Agree/Strongly Agree (*n*) %	Unsure (*n*) %	Disagree/Strongly Disagree (*n*) %
**PSYCHOEDUCATION** **In this program I learned…**			
What biofeedback is	(52) 88.1	(7) 11.9	(0) 0.0
About the link between pain and stress	(57) 96.6	(2) 3.4	(0) 0.0
How thoughts and feelings can make pain better or worse	(59) 100.0	(0) 0.0	(0) 0.0
How to identify symptoms of anxiety and depression	(46) 78.0	(11) 18.6	(2) 3.4
Why it’s important to get more help if I have symptoms of anxiety and depression	(52) 88.1	(5) 8.5	2 (3.4)
**PAIN NEUROSCIENCE EDUCATION** **In this program I learned…**			
About how pain works in the body	(57) 96.6	(0) 0.0	(2) 3.4
About the mind-body connection	(57) 98.3	(0) 0.0	(1) 1.7
About the importance of staying active and engaged in my life	(57) 96.6	(2) 3.4	(0) 0.0
**SKILLS TRAINING**			
I learned how to do diaphragmatic breathing/belly breathing	(56) 94.9	2 (3.4)	(1) 1.7
I know how to practice belly breathing on my own	(53) 91.4	(2) 3.4	(3) 5.2
I learned how to do guided imagery	(55) 93.2	(4) 6.8	(0) 0.0
I know how to practice guided imagery on my own			
I learned how active distractions (like art) can help to reduce my focus on pain	(56) 94.9	(2) 3.4	(1) 1.7
I developed my own Comfort Ability^®^ plan	(54) 93.1	(3) 5.2	(1) 1.7
I enjoyed the activities	(54) 91.5	(4) 6.8	(1) 1.7
**SOCIAL SUPPORT**			
I was able to ask questions and talk about things that were important to me	(55) 94.8	(2) 3.4	(1) 1.7
I found it helpful to talk to other kids who have pain	(53) 91.4	(5) 8.6	(0) 0.0
There was enough time for asking questions and talking	(57) 96.6	(1) 1.7	(1) 1.7
**GENERAL INTERVENTION RESPONSE**			
I have more confidence in managing my pain	(46) 79.3	(9) 15.5	(3) 5.2
I will make changes to my routines that may help me to feel better	(55) 94.8	(2) 3.4	(1) 1.7
I wish there was a follow up program for more support	(36) 62.0	(15) 25.9	(7) 12.1
I am glad I did this program	(53) 91.4	(5) 8.6	(0) 0.0

**Table 3 children-10-01523-t003:** Caregiver Post-Treatment Satisfaction Survey.

	Agree/Strongly Agree (*n*) %	Unsure (*n*) %	Disagree/Strongly Disagree (*n*) %
**PSYCHOEDUCATION** **In this program I learned:**			
About cognitive behavioral therapy	(86) 95.6	(2) 2.2	(2) 2.2
About the relationship between pain, stress, negative feelings, and worry	(86) 94.5	(4) 4.4	(1) 1.1
How relaxation skills can help to ease pain	(88) 97.8	(1) 1.1	(1) 1.1
**PAIN NEUROSCIENCE EDUCATION** **In this program I learned:**			
How pain functions in the body (pain science)	(88) 96.7	(1) 1.1	(2) 2.2
The difference between acute and chronic pain	(89) 97.8	(0) 0.0	(2) 2.2
How age and development impact a child’s pain	(71) 78.0	(18) 19.8	(2) 2.2
**SKILLS TRAINING** **In this program I learned:**			
New ideas for how to respond to my child’s pain (1st vs. 2nd intuitions)	(87) 95.6	(1) 1.1	(3) 3.3
How to use reflective listening to communicate better with my child	(86) 94.5	(4) 4.4	(1) 1.1
About the importance of keeping focus on function to best support my child	(86) 96.6	(1) 1.1	(2) 2.2
How to make a scaffolding plan for school, sleep, exercise, or chores	(87) 95.6	(1) 1.1	(3) 3.3
How to do basic relaxation skills like breathing and mindfulness	(90) 98.9	(0) 0.0	(1) 1.1
About the importance of self-care	(89) 97.8	(1) 1.1	(1) 1.1
**SOCIAL SUPPORT**			
There was enough time to ask questions and discuss things that were important to me	(74) 83.1	(7) 7.9	(8) 9.0
The time spent talking and sharing with other parents was useful	(79) 86.8	(7) 7.7	(5) 5.5
**GENERAL INTERVENTION RESPONSE**			
After this program, I will be making changes in how I handle my child’s pain	(86) 94.5	(2) 2.2	(3) 3.3
I would be interested in more programs like this	(81) 89.0	(8) 8.8	(2) 2.2

## Data Availability

The data presented in this study are available upon request from the corresponding author. The data are not publicly available because this is a preliminary investigation. When full data are published, they will be made publicly available.

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
