# Peer review of "Rapid Mobilization of an Evidence-Based Psychological Intervention for Pediatric Pain during COVID-19: The Development and Deployment of the Comfort Ability® Program Virtual Intervention (CAP-V)"

_children, 2023, doi:10.3390/children10091523_

Round 1

Reviewer 1 Report

Dear Authors,

Scientific soundness of the paper is amazing. 

Good luck with further publication!

Author Response

Thank you for your kind words.  We appreciate your feedback!

Reviewer 2 Report

Review:

Rapid Mobilization of an Evidence-Based Psychological Intervention for Pediatric Pain During COVID-19: The Development and Deployment of the Comfort Ability® Program Virtual Intervention (CAP-V).

The authors of the manuscript did an excellent job of explaining the program test was designed the program test was designed to address pain in children. If this manuscript is published their approach can be widely employed in hospitals, reducing the burden on healthcare, and improving the quality of life of children suffering from chronic pain.

Author Response

Thank you so much for your review and for your enthusiasm for this work.  

Reviewer 3 Report

line 226 - don't need "and" before "caregiver's marital status"

lines 286-298 - needs more clarity. What does the 39.2% refer to?

line 487 - should be "time points" not "time point"

line 491 - "COVD-19" should be "COVID-19"

line 536 - "Ottowa" is not the correct spelling for "Ottawa"

line 546 - capitalize the first letter each word in journal name 

Overall, great use of KTA framework for rapidly deploying a much needed program during the pandemic, which not only helped fill a gap at that time but now also provides an alternative for pediatric chronic pain patients and their caregivers on an ongoing basis. Looking forward to the studies in-progress that are an extension of this paper. I am hopeful they will help address some of the limitations you've highlighted and will provide more robust continent-wide data.

Author Response

Thank you so much for your kind review.  We appreciate your attention to detail and we have corrected all of these typos in the final draft. 

Reviewer 4 Report

This manuscript entitled Rapid Mobilization of an Evidence-Based Psychological Intervention for Pediatric Pain During COVID-19: The Development and Deployment of the Comfort Ability® Program Virtual Intervention (CAP-V) describes the dissemination of an important intervention via virtual delivery during the pandemic when the in-person option to deliver CAP was unavailable. This is a well written and described manuscript that offers readers a methodology KTA (Knowledge-To-Action) framework showcasing ways evidence-based interventions can be modified and delivered in a timely way ensuring patients and their families have access to evidence-based treatment.

I don’t see any weaknesses outside of those outlined in the limitations section by the authors. All of the sections are well described with important details that would allow for replication on the procedures and validation of the results. If this is not too challenging to add-I would recommend that a comparative analysis be attempted by looking at perhaps zip codes to highlight the distance that families who attended CAP-V saved in travel compared to those who attended CAP-IP. The resources that family save in gas, time off from work, sometimes even having to stay in a local hotel etc when they travel for CAP-IP make it challenging for some to access this valuable treatment. Highlighting those savings when we improve access with interventions delivered virtually such as CAP-V would greatly enhance this already very strong manuscript (totally optional).

Minor edit-

Line 330- delete the word “they”

Author Response

Thank you so much for your helpful review. We love the idea of doing a zip code comparison between CAP-V and CAP-IP to illustrate potential differences in healthcare utilization between the two workshops.  However, we do not have access to that data in our current study parameters (it would require a new IRB approval for chart review) and attaining this data would significantly delay this publication.  We agree that this data would be helpful in justifying ongoing virtual resources and will try to incorporate this into our future work.  Many thanks!